# Comparison of Liquefied Petroleum Gas Cookstoves and Wood Cooking Fires on PM_2.5_ Trends in Brick Workers’ Homes in Nepal

**DOI:** 10.3390/ijerph17165681

**Published:** 2020-08-06

**Authors:** James D. Johnston, Megan E. Hawks, Haley B. Johnston, Laurel A. Johnson, John D. Beard

**Affiliations:** 1Department of Public Health, Brigham Young University, Provo, UT 84602, USA; hawks.megan@gmail.com (M.E.H.); john_beard@byu.edu (J.D.B.); 2Department of Plant and Wildlife Sciences, Brigham Young University, Provo, UT 84602, USA; haleyxc4@gmail.com; 3Marriott School of Business, Marketing & Global Supply Chain, Brigham Young University, Provo, UT 84602, USA; laureljohnson01@gmail.com

**Keywords:** household air pollution, fine particulate matter, brick workers, indoor environmental quality, international environmental health, cookstove

## Abstract

Prior studies document a high prevalence of respiratory symptoms among brick workers in Nepal, which may be partially caused by non-occupational exposure to fine particulate matter (PM_2.5_) from cooking. In this study, we compared PM_2.5_ levels and 24 h trends in brick workers’ homes that used wood or liquefied petroleum gas (LPG) cooking fuel. PM_2.5_ filter-based and real-time nephelometer data were collected for approximately 24 h in homes and outdoors. PM_2.5_ was significantly associated with fuel type and location (*p* < 0.0001). Pairwise comparisons found significant differences between gas, indoor (geometric mean (GM): 79.32 μg/m^3^), and wood, indoor (GM: 541.14 μg/m^3^; *p* = 0.0002), and between wood, indoor, and outdoor (GM: 48.38 μg/m^3^; *p* = 0.0006) but not between gas, indoor, and outdoor (*p* = 0.56). For wood fuel homes, exposure peaks coincided with mealtimes. For LPG fuel homes, indoor levels may be explained by infiltration of ambient air pollution. In both wood and LPG fuel homes, PM_2.5_ levels exceeded the 24 h limit (25.0 µg/m^3^) proposed by the World Health Organization. Our findings suggest that increasing the adoption of LPG cookstoves and decreasing ambient air pollution in the Kathmandu valley will significantly lower daily PM_2.5_ exposures of brick workers and their families.

## 1. Introduction

Household air pollution from indoor burning of solid biomass and gaseous fuels is responsible for approximately 3.8 million deaths annually worldwide [1]. Indoor burning of solid biomass fuels, including wood, on open fires produces high levels of air pollutants, including inhalable (PM_10_) and fine (PM_2.5_) particulate matter, carbon monoxide (CO), nitrogen oxides (NO_x_), and an array of chemicals that are toxic or carcinogenic to humans [2,3,4,5,6]. Chronic exposure to household air pollution is associated with acute lower respiratory tract infections, respiratory illness, impaired immune function, and low birth weight in children, and chronic obstructive pulmonary disease (COPD), respiratory illnesses, tuberculosis (TB), impaired immune function, cardiovascular disease, cataracts, and lung cancer in adults [5,6,7]. Efforts to reduce exposure to household air pollution by providing improved stoves to exposed populations have been met with mixed results, largely due to cultural barriers, cost, and local social and environmental factors [8,9,10,11]. Thus, understanding population-specific exposure patterns, fuel use, and cooking and heating practices is a necessary first step in developing tailored, culturally appropriate interventions to reduce exposure. 

Brick manufacturing is a major industry in Nepal and a potentially large source of household air pollution exposure within the country. Estimates of the number of operating brick kilns in Nepal range from 429 to 750 [12,13,14], with a workforce estimate of up to 400,000 seasonal workers annually [13,15]. In the Kathmandu valley alone, there are over 100 brick kilns that employ approximately 30,000 workers in total [16,17,18]. Brick workers in Nepal experience high rates of respiratory symptoms, including chronic cough, chronic phlegm, chronic bronchitis, wheezing, asthma, dyspnea, and chest tightness [19]. These respiratory symptoms may be partially due to occupational exposures, particularly for dusty tasks such as moving red bricks from the kiln after they are fired [20]. Brick workers’ exposures to silica and other hazardous dusts are well documented globally [21,22,23]. In Nepal, it is common for brick workers to live in impoverished conditions, often on-site at the brick kiln. Brick workers and their families typically live in rudimentary huts composed of brick walls and a tin roof, with an average floor area of less than 8.0 m^2^ [24]. During mealtimes, some brick workers and their families rely on wood fuel for cooking, which may emit dangerous levels of particulate matter and other pollutants inside the home. Thus, household air pollution exposure during non-working hours may compound brick workers’ occupational exposures and may partially explain the high rates of respiratory symptoms in this population.

To date, there are few studies on indoor air pollution exposure among brick workers and their families globally or specifically in Nepal. One previous study evaluated indoor PM_2.5_ exposure in brick workers’ homes in Bhaktapur [24]. However, exposure monitoring in that study occurred during the middle of the day for approximately seven hours, during which time most of the residents were not home. As a result, PM_2.5_ levels were not measured during peak cooking hours, and results showed no significant difference between indoor and outdoor concentrations [24]. The two primary cooking methods used among brick workers and their families in Nepal are cook stoves using liquefied petroleum gas (LPG) and open fires using wood. Given the relatively low cost and global availability of LPG, the World Health Organization (WHO) has stated that studies of the impact of LPG on reducing household air pollution exposure are needed as soon as possible [25]. Therefore, we aimed in this study to characterize indoor PM_2.5_ levels in LPG vs. wood fuel homes of brick workers and their families over a 24 h period. Understanding exposure patterns related to fuel use may help to guide future interventions to reduce PM_2.5_ exposure and respiratory symptoms in this vulnerable population of workers and their families. 

## 2. Materials and Methods 

### 2.1. Study Design

We used a cross-sectional study design to compare indoor PM_2.5_ levels, air temperature (°C), and relative humidity (RH; %) in brick workers’ homes according to the primary fuel used for cooking. Homes were classified as either LPG cookstove or woodfire homes. Figure 1 shows the typical construction of homes sampled in this study. Figure 2 illustrates LPG and wood fire cooking arrangements observed in this study. Homes (N = 19) were selected by convenience sampling at a single brick kiln in Bhaktapur, Nepal. All study homes were located at the worksite, within an approximately 300 m radius of the brick kiln. Samples were collected from 30 April to 3 May 2019. Measurements were collected by placing the monitoring devices on a tripod positioned at 1.2 m from the floor, which we estimated to be the approximate breathing zone height of an adult crouching in the home. For comparison, we simultaneously collected outdoor PM_2.5_, air temperature, and RH measurements away from observed emission sources and within 200 m of the study homes. A brief housing survey was also administered to an adult occupant of the home by means of an interpreter. Brigham Young University’s Institutional Review Board (IRB) reviewed this study. This study did not meet the definition of human subjects research outlined in 45 CFR 46 [26] because the unit of study was the home rather than the individual and, therefore, it was determined that the study did not require IRB approval. 

### 2.2. Indoor PM_2.5_ Measurement

All PM_2.5_ samples were collected using RTI International’s MicroPEMs V 3.2A (RTI International, Research Triangle Park, NC, USA). The MicroPEMs allow for both gravimetric (filter-based) sampling and real-time datalogging using an on-board nephelometer. The MicroPEM nephelometers were set to record PM_2.5_ concentrations every 10 s over the sampling period. Gravimetric PM_2.5_ samples were collected on 3.0 μm PTFE 25 mm filters (Zefon International, Ocala, FL, USA). We pre-weighed filters using a Mettler Toledo (Mettler Toledo, Columbus, OH, USA) model XP2U microbalance in a temperature and humidity-controlled room. Filters were conditioned in the room for 24 h prior to weighing. We used Teflon™ coated tweezers (Mettler Toledo) to handle filters. We used a Haug (Haug North America, Williamsville, NY, USA) U-bar deionizer to remove static electricity from each filter prior to weighing. We weighed and deionized each filter 3 times. The mean of these weights was recorded as the pre-weight value. Filters were then placed in SKC filter keepers (SKC, Inc., Eighty Four, PA, USA) until use. The same procedure was followed after returning from Nepal for determining post-weights.

Prior to sample collection each day, we used Docking Station software version 2.0 (RTI International, Research Triangle Park, NC, USA) to interface with the MicroPEMs to set the instrument date/time, flow rate, and nephelometer offset. Nephelometer offsets were adjusted by placing a high efficiency particulate air (HEPA) filter in-line before the flowmeter to provide particle free air. The offset for the 780 nm IR laser was adjusted until the particle count was zero. A TSI model 4140 mass flowmeter (TSI, Shoreview, MN, USA) was used to set the MicroPEM flowrate at 0.50 L/min. Sintered stainless steel impaction plates were cleaned and oiled by RTI International prior to data collection. MicroPEMs were attached to tripods by means of a pouch made of low-dander ripstop nylon. After sample collection, nephelometer data were downloaded from the MicroPEMs using Docking Station. MicroPEM flow rates were checked after sampling, and all instruments were found to have minimal drift of less than ± 5% from 0.50 L/min and were thus determined to be valid samples [27]. Finally, the memory was cleared from each instrument before preparing the instrument for the next sampling event. For all homes sampled, the median sampling time was 21.21 h (interquartile range = 2.04 h).

### 2.3. Indoor Temperature and RH Measurement

Air temperature and RH data were collected in each home using Extech SD500 dataloggers (Extech Instruments, Nashua, NH, USA). Prior to data collection each day, we installed fresh batteries, cleared the SD card, set the date and time, and set the logging interval to record measurements every five minutes. Following the sampling period, data were downloaded from the instruments and saved as Excel spreadsheets. For all homes, the median sampling time was 21.42 h (interquartile range = 1.83 h). 

### 2.4. Outdoor PM_2.5_, Temperature, and RH Measurement

For comparison with the indoor samples, we collected daily outdoor PM_2.5_, temperature, and RH measurements in a centralized location at the brick kiln, away from any observed pollution sources. Outdoor samples were collected under a covered pavilion on the kiln property. The pavilion roof was approximately 8 m high, with open sides, allowing for natural air flow. An RTI MicroPEM and Extech SD500 datalogger were positioned at 1.2 m from the ground on a tripod under the pavilion. For outdoor samples across all days, the median sampling time for the MicroPEMs was 21.98 h, and the mean and standard deviation (SD) for the Extech SD500 dataloggers were 20.62 and 1.31 h, respectively. We handled all instruments and data with the same methods as those described for the indoor samples.

### 2.5. Housing Questionnaire

Housing factors were assessed using an extant 14-item questionnaire [24] that was modified for use in this study. Specifically, we limited the number of questions to five, including number of people living in the home, number of children living in the home, primary fuel used for cooking, smokers in the home, and, if so, the number of smokers. Fuel and cooking device were verified by visual inspection of the home. While questionnaires were administered, study personnel measured the living area (m^2^) of the home. Occupant density was calculated as the number of people living in the home divided by the living area. Study personnel filled out the questionnaires by interviewing an adult occupant of each home with the help of an interpreter. 

### 2.6. Statistical Analyses

Prior to conducting statistical analyses, we corrected the MicroPEM nephelometer readings based on the gravimetric results from the on-board filter, as recommended by RTI International [28]. The correction factor was calculated as the filter concentration divided by the mean nephelometer concentration over the sampling period. We then multiplied all of the MicroPEM nephelometer PM_2.5_ data by the correction factor.

We conducted all statistical analyses using SAS version 9.4 (SAS Institute Inc., Cary, NC, USA) and used a significance level of α = 0.05. We calculated frequencies and percentages for categorical brick kiln home characteristics and the arithmetic mean, standard deviation, minimum, first quartile, median, third quartile, and maximum for continuous home characteristics and for mean (over the sampling period) PM_2.5_ concentration, RH, and temperature.

We used simple linear regression models to estimate unadjusted associations, with 95% confidence intervals, between brick kiln home characteristics and three primary outcome variables: mean (over the sampling period) PM_2.5_ concentration, RH, and temperature. The distribution of mean PM_2.5_ concentration was right-skewed, so we natural logarithm transformed the mean PM_2.5_ concentrations before including that variable as the outcome in linear regression models and then exponentiated the regression coefficients. We evaluated pairwise differences in the outcome variables among categories of fuel type and location and used the Tukey method to adjust *p*-values for multiple comparisons. Additionally, we made box plots that showed mean PM_2.5_ concentration, RH, and temperature according to fuel type and location. We also fit some multiple linear regression models that contained two home characteristics as independent variables.

To assess the diurnal variation of PM_2.5_ concentration, RH, and temperature, we made line graphs of mean (over fuel type and location categories) PM_2.5_ concentration, RH, and temperature over time.

## 3. Results

Samples were collected over a three-day period, during which we sampled 19 homes and collected daily outdoor measurements from a central location at a single brick kiln in Bhaktapur, Nepal. The median home area was 10.25 m^2^, mean number of people in the home was 3.42, and median occupant density was 29.70 people per 100 m^2^ (Table 1). Fifty-three percent of homes had 0–1 child in the home, while 2–3 children were present in 47% of homes. Smokers were present in 53% of homes and the mean number of smokers in the home was 0.76. Sixty-seven percent of homes used LPG, while 33% used wood as the primary fuel for cooking.

The median average (over the sampling period) PM_2.5_ concentration for all samples combined was 118.46 μg/m^3^ (Table 2). The mean average RH was 57.87% and the mean average temperature was 24.58 °C for all samples combined.

For the mean (over the sampling period) PM_2.5_ concentration, there were significant associations with smokers in the home (yes: geometric mean (GM) = 301.35 μg/m^3^; no: GM = 61.65 μg/m^3^; *p* = 0.0005), number of smokers in the home (exponentiated regression coefficient = 1.74; *p* = 0.03), and fuel type and location (*p* < 0.0001) (Table 3). Pairwise comparisons found significant differences between gas, indoor (GM = 79.32 μg/m^3^), and wood, indoor (GM = 541.14 μg/m^3^; *p* = 0.0002), and between wood, indoor, and outdoor (GM = 48.38 μg/m^3^; *p* = 0.0006) but not between gas, indoor, and outdoor (*p* = 0.56). For mean RH, there was a significant association with fuel type and location (*p* < 0.0001). Pairwise comparisons found significant differences between gas, indoor (mean = 54.40%), and outdoor (mean = 67.97%; *p* < 0.0001) and between wood, indoor (mean = 58.51%), and outdoor (*p* = 0.006) but not between gas, indoor, and wood, indoor (*p* = 0.15). Smokers in the home (yes: mean = 25.63 °C; no: mean = 24.58 °C; *p* = 0.05), number of smokers in the home (regression coefficient = 0.55 °C; *p* = 0.04), and fuel type and location (*p* < 0.0001) were significantly associated with mean temperature. Pairwise comparisons found significant differences between gas, indoor (mean = 24.67 °C) and wood, indoor (mean = 26.20 °C; *p* = 0.02), between gas, indoor, and outdoor (mean = 22.11 °C; *p* = 0.001), and between wood, indoor, and outdoor (*p* < 0.0001). No other home characteristic was significantly associated with mean PM_2.5_ concentration, RH, or temperature.

Box plots for mean PM_2.5_ concentration, RH, and temperature according to fuel type and location showed patterns that were similar to those just described (e.g., the distribution of mean PM_2.5_ concentration was higher for wood, indoor than for gas, indoor, and outdoor) (Figure 3).

Including smokers in the home and fuel type and location as independent variables in the multiple linear regression model for mean PM_2.5_ concentration did not change the significance of associations (smokers in the home: *p* = 0.04; fuel type and location: *p* = 0.03). However, fuel type and location (*p* = 0.008) remained significantly associated with mean PM_2.5_ concentration, but number of smokers in the home did not (*p* = 0.87), when number of smokers in the home and fuel type and location were included as independent variables in the multiple linear regression model for mean PM_2.5_ concentration. Fuel type and location (*p* = 0.03) remained significantly associated with mean temperature, but smokers in the home did not (*p* = 0.63), when smokers in the home and fuel type and location were included as independent variables in the multiple linear regression model for mean temperature. Similarly, fuel type and location (*p* = 0.04) remained significantly associated with mean temperature, but number of smokers in the home did not (*p* = 0.69), when number of smokers in the home and fuel type and location were included as independent variables in the multiple linear regression model for mean temperature.

The line graph of mean (over fuel type and location categories) PM_2.5_ concentration over time showed increases in mean PM_2.5_ concentrations primarily at mealtimes, particularly in homes that burned wood indoors (Figure 4). Mean RH decreased until mid-afternoon, then increased until the morning, and then decreased again, particularly outdoors. The pattern for mean temperature was opposite to the pattern for mean RH.

## 4. Discussion

Indoor cooking and heating with biomass fuels on open fires, as opposed to cleaner biomass burning devices such as chimney cookstoves and gasifiers, is recognized as a major source of PM_2.5_ exposure globally, particularly among populations living below the poverty level [25]. Prior studies in Nepal report high levels of household air pollutants due to biomass fuel use among various populations [29,30,31]. However, there is little data specific to brick kiln workers, who represent a uniquely vulnerable and relatively large workforce in the Kathmandu valley. Brick kiln workers are at increased risk for respiratory diseases due to crowded housing [24], low socio-economic status, and occupational exposures associated with brick manufacturing [19,32]. Adding to these risks, the results presented in this study suggest that brick kiln workers and their families are potentially exposed to dangerously high levels of PM_2.5_ during non-working hours. Specifically, GM PM_2.5_ levels in homes using wood and LPG as the primary cooking fuels were 21.6 and 3.2 times the WHO’s recommended 24 h limit of 25.0 µg/m^3^ [25], respectively. Considering the high prevalence of respiratory symptoms in this population, household air pollution exposures should be considered as a potentially significant environmental risk factor, warranting additional research on the topic. 

Our findings indicate that PM_2.5_ concentration was significantly associated with device-fuel type (open wood fire vs. LPG cookstove). This finding differs from PM_2.5_ levels reported by Thygerson et al. (2019) showing that differences by fuel type were not significant [24]. Of note, however, air samples collected by Thygerson et al. were collected during the middle of the workday and did not capture exposures during morning and evening mealtimes. Thus, their results may mostly reflect the infiltration of outdoor air pollution into homes. One advantage of this study is that we collected samples across an approximately 24 h period, capturing mealtime cooking. We also simultaneously sampled PM_2.5_ using optical and gravimetric methods, allowing us to identify trends in concentration over time. Using this sampling strategy, we were able to show that increases in mean PM_2.5_ concentration largely appear to correspond with mealtimes, particularly for wood fuel homes. 

Although PM_2.5_ levels in LPG homes in this study were still significantly higher than the WHO’s guidelines, they were much lower than levels in wood fuel homes. A sizeable proportion of the indoor PM_2.5_ pollution in LPG homes likely originates from ambient air pollution infiltrating into homes. Study homes were mostly made of un-mortared brick walls, with gaps between the bricks, the roof and walls, and at the door opening. Thygerson et al. (2019) compared PM_2.5_ pollution in brick workers’ homes at four kilns in Bhaktapur, Nepal, and found that the overall PM_2.5_ concentration and the elemental and carbon components did not differ between indoor and outdoor samples. Indeed, the indoor/outdoor (I/O) ratio for PM_2.5_ was 0.98, suggesting that gaps in building materials for these homes allow for significant infiltration of outdoor air. Most brick-making tasks occur outside, where workers are exposed to daily ambient air pollution while simultaneously being exposed to occupational dust and silica [20,32]. Based on our results, brick kiln workers do not appear to have a recovery period from hazardous inhalation exposure, as indoor PM_2.5_ levels remained above the WHO’s guidelines overnight, even in LPG homes. In addition to providing LPG fuel and stoves for brick workers (as opposed to wood or other biofuels), efforts to decrease PM_2.5_ exposure in this population must also focus on lowering ambient air pollution levels in the Kathmandu valley, which have increased significantly in recent years [30,33,34,35]. 

We are not aware of any formal assessment of household fuel use among brick workers in Nepal. Anecdotally, it appeared that more homes used LPG fuel than wood for cooking at the kiln included in this study. However, approximately 60% of the homes in the study by Thygerson et al. (2019) used wood fuel [24], suggesting that fuel use patterns may vary by kiln. The WHO recommends that, prior to policy planning, household energy use (fuels used for cooking, heating, and lighting) should be assessed for specific populations [25]. Based on findings from this study and that of Thygerson et al., we agree that a fuel use assessment in brick workers’ housing is needed to inform future intervention measures. LPG gas stoves may already be adopted by a relatively large proportion of brick workers and may provide a cleaner alternative to wood fuel. However, based on the WHO’s guidelines, a formal evaluation is needed to determine the cultural acceptability, sustainability, cost, supply, and potential exposures and safety to home occupants of transitioning to LPG cooking fuel [25]. 

In the absence of large fossil fuel sources, Nepal is obliged to import LPG fuel primarily from India [36]. The demand for LPG fuel, particularly in urban areas such as the Kathmandu valley, is expected to increase by 3.9%–4.9% per year through 2040 [37], even while the cost of LPG has increased by 8% annually [38]. Currently, the government of Nepal subsidizes the cost of LPG, but there is a movement to discontinue these subsidies and to divert that money to the development of hydroelectric power [36]. Thus, efforts to transition brick workers from open wood fires to LPG cookstoves will require monitoring and reporting to identify ongoing sociopolitical impacts, such as increasing LPG fuel cost, that may create barriers to use. Given the vast, undeveloped hydroelectric potential in Nepal [36], low-cost electric cookstoves may provide a long-term solution to decrease indoor air pollution exposure among brick workers. However, there may also be a significant time lag for the development of hydroelectric power generation and distribution of sufficient electricity throughout the Kathmandu valley. This may result in ongoing, long-term exposure to household air pollution among brick workers that could, in the interim, be lowered by the temporary use of LPG cookstoves. In addition, cooking stacking (using multiple fuel/device options) may be an obstacle to cleaner energy use in some populations [39]. Providing brick workers with more than one clean fuel-device option may help to prevent reversion to pollution prone options such as open wood fires. A clearer understanding of fuel use patterns, and barriers to adopting LPG fuel, is an essential first step to guide interventions to decrease household air pollution exposure among brick workers in Nepal, with electric cookstoves or clean cooking stacking options as potential long-term solutions. 

A previous study conducted on the prevalence of respiratory illnesses among brick workers in Nepal shows that smoking was a significant predictor (*p* < 0.001) of chronic bronchitis and chronic wheezing [19]. Of the world’s 1.1 billion global smokers, around 80% live in low- and middle-income countries [40]. In Nepal, 27.4% of adult males and 5.5% of adult females smoke tobacco, and the prevalence of smoking is higher among those with lower levels of education [41]. In our study, approximately half of the homes sampled had at least one smoker living in the home. This is similar to the findings of Thygerson et al. (2019). In their study, 67% of homes had a least one smoker, many of which reported smoking indoors [24]. We also found that PM_2.5_ levels were significantly higher in homes with smokers compared to homes with no smokers. Thus, we recognize that multi-faceted interventions that include smoking cessation are necessary in order to decrease respiratory illness among brick workers in Nepal. The primary national intervention strategies to limit tobacco use in Nepal are prescribed in the Tobacco Product (Control and Regulatory) Act, which was signed into law in 2011. This law requires the size of warning labels on tobacco products to cover at least 90% of the package area, prohibits the distribution and advertisement of tobacco products to children, levies a tax on tobacco products, and bans smoking in public areas, among other provisions [42,43]. Worksite smoking cessation interventions have shown some success in developed countries [44,45]. In addition to the national intervention measures, public health agencies and/or non-governmental organizations (NGOs) could work directly with individual brick kilns to implement worksite smoking cessation programs that include financial incentives and other evidence-based strategies [44,45]. 

Our findings regarding temperature and RH may be cause for further concern regarding brick workers’ respiratory and overall health. The average RH in brick workers’ homes was approximately 58%. As evenings dropped to cooler temperatures, RH in both gas and wood burning homes increased to levels between 55% and 75% and remained within this range for several hours, often until almost noon on the next day. This is a concern because house dust mites (HDMs) grow at RH levels between 55% and 75% [46,47]. HDM allergens are among the most clinically significant antigen exposures affecting humans [48,49,50] and exposure to them is associated with the development of asthma and other allergic diseases [48,49,50]. In light of the already present risks of ambient and indoor air pollution, the possibility of HDMs within brick workers’ bedding presents an added health concern. We recommend further study investigating the presence of HDMs as a possible contributing factor to the high levels of respiratory problems among brick kiln workers in Nepal [19,32].

To our knowledge, this is the first study to characterize indoor PM_2.5_ levels in brick workers’ homes in Nepal for an approximately 24 h period. One strength of our study design is that we simultaneously collected filter-based (gravimetric) and optical particle counts, allowing us to identify PM_2.5_ trends over time. This was advantageous in that it allowed us not only to identify mean differences between wood and LPG fuel homes but also to see the influence that cooking has on indoor pollution levels, which was not studied in previous research by Thygerson et al., 2019 [24]. 

### Limitations

This study was limited to a relatively small number of homes at a single brick kiln in Bhaktapur, Nepal. Thus, our study may have been underpowered to detect some associations (i.e., type II error) between brick kiln home characteristics and the three outcomes and our point estimates had wide 95% CI. It is also possible that our small sample size could have led to spurious results (i.e., type I error). In addition, we used a convenience sample of homes and our findings may not be generalizable to other kilns in the Kathmandu valley or to other clay brick kilns worldwide. In addition, we were limited to sampling brick workers’ homes over one 24 h period during a single season, which also may have contributed to lower statistical power and wider 95% CI than if we had been able to collect multiple 24 h samples from each home. Longer-term sampling or repeated sampling of the same homes across seasons will likely provide a much better understanding of PM_2.5_, temperature, and RH levels in brick workers’ homes. Unmeasured confounding by temporal factors could have affected associations between brick kiln home characteristics and the three outcomes. We did not collect information regarding technologies (e.g., open fires vs. cookstoves that had venting/chimneys) and, therefore, could not estimate associations between use of technologies and PM_2.5_, RH, and temperature levels or the combined effect of use of fuels and technologies on these outcomes. Finally, this study was limited to environmental measurements. Considering the relatively high levels of indoor PM_2.5_ in both LPG and wood fuel homes, studies of respiratory and other health effects associated with brick workers’ housing are needed. 

## 5. Conclusions

In this study, indoor PM_2.5_ levels for both wood and LPG fuel homes exceeded the WHO’s recommended 24 h limit of 25.0 µg/m^3^, but the exceedance was most pronounced for wood fuel homes. Peaks in PM_2.5_ appeared to coincide largely with mealtimes in wood fuel homes. For LPG homes, mealtime peaks were significantly less pronounced, and overall levels may be significantly influenced by the infiltration of ambient air pollution. Based on our findings and those of a previous study [24], LPG cookstoves appear to be used by a relatively large proportion of brick workers, suggesting that they may be a culturally appropriate alternative to wood fuel cooking. Future studies should focus on understanding barriers to LPG stove use, the potential for the adoption of clean cooking stacking, as well as interventions to decrease the use of wood fuel for cooking in this population of workers.

## Figures and Tables

**Figure 1 ijerph-17-05681-f001:**
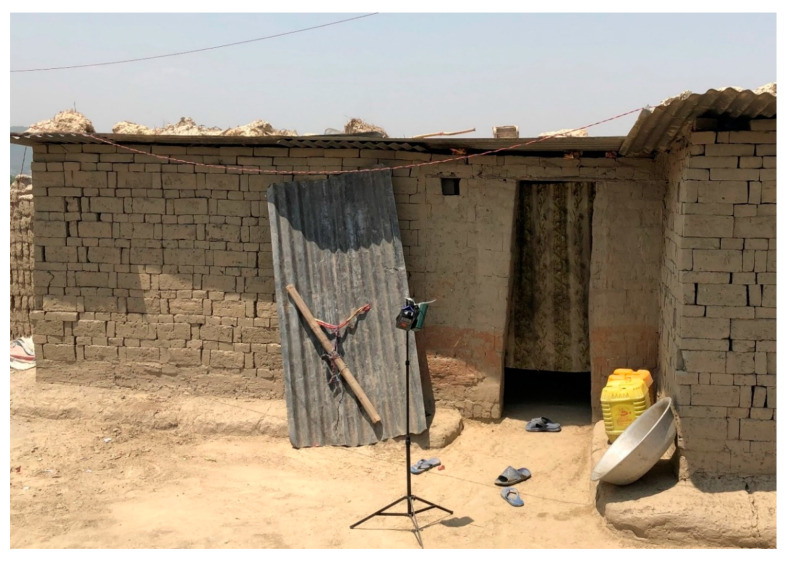
Typical construction of brick workers’ homes sampled in this study. Construction consisted of dirt floors and un-mortared brick walls. Tin roofs were constructed by placing bamboo or wooden poles across the top of the walls, with sheet metal placed on the poles. Rocks, bricks, soil, and other heavy objects were used to hold the sheet metal in place. Indoor samples were collected by attaching PM_2.5_ and temperature/relative humidity instruments to tripods as shown. Tripods with the attached instruments were placed inside the home during the entire sampling period. Abbreviation: PM_2.5_, particulate matter with an aerodynamic diameter less than 2.5 μm.

**Figure 2 ijerph-17-05681-f002:**
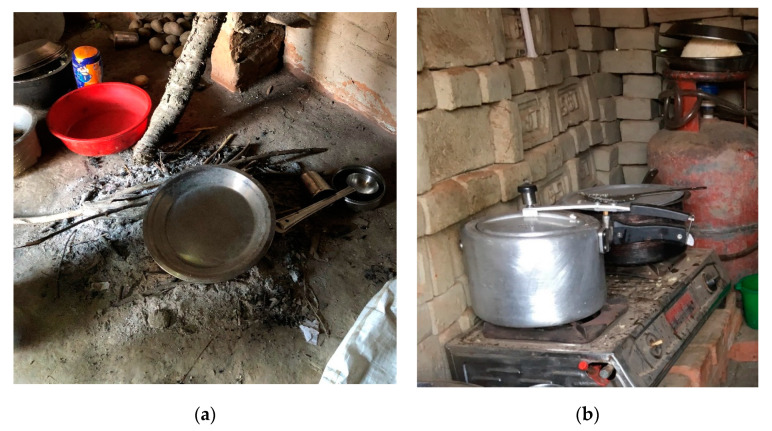
Illustrative examples of the two cooking practices observed in brick workers’ homes in Bhaktapur, Nepal: (**a**) indoor wood fire and (**b**) LPG (liquefied petroleum gas) cook stove.

**Figure 3 ijerph-17-05681-f003:**
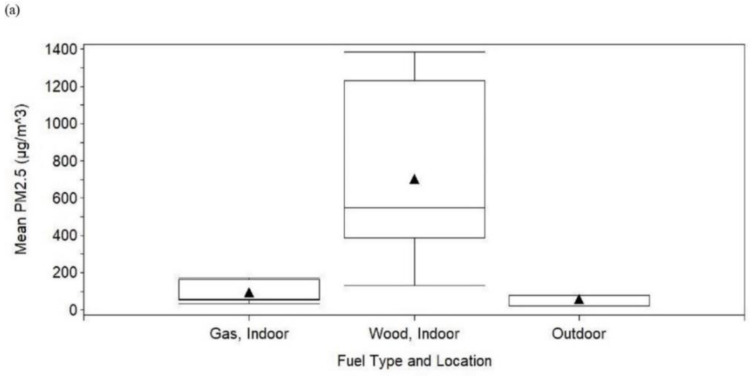
Box plots of the mean of samples inside or outside ^a^ homes by fuel type and location at a brick kiln in Bhaktapur, Nepal, May 2019: (**a**) PM_2.5_, (**b**) relative humidity, (**c**) temperature. Reading from the top to the bottom, the horizontal lines on each box plot represent the maximum, third quartile, median, first quartile, and minimum, and the black triangle represents the mean. Abbreviation: PM_2.5_, particulate matter with an aerodynamic diameter less than 2.5 μm. ^a^ Nineteen samples were from inside and four samples were from outside the homes (for PM_2.5_, 17 samples were from inside and three samples were from outside the homes).

**Figure 4 ijerph-17-05681-f004:**
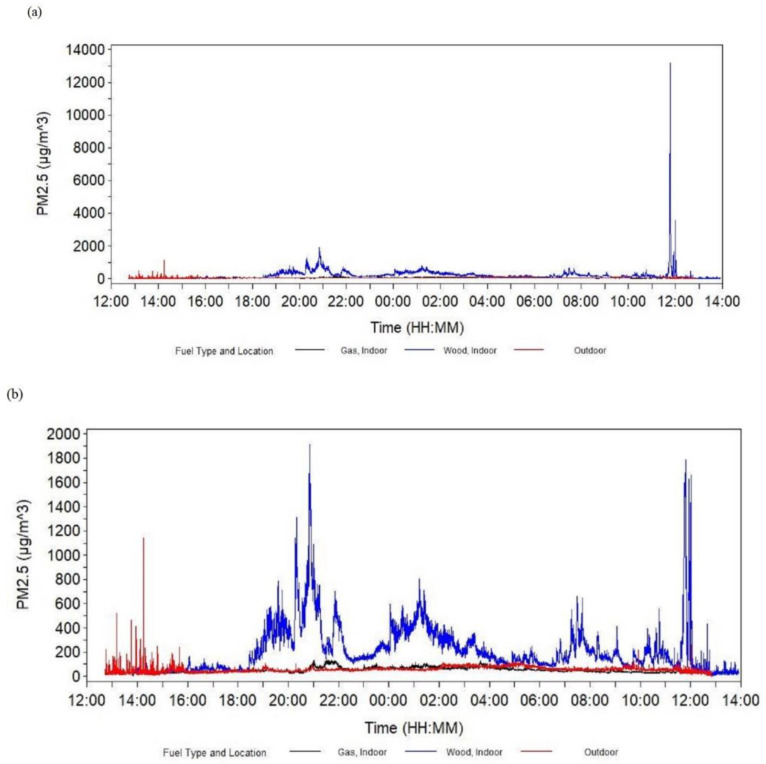
Line graphs of the mean of samples inside or outside ^a^ homes over time by fuel type and location at a brick kiln in Bhaktapur, Nepal, May 2019: (**a**) PM_2.5_, (**b**) subset of PM_2.5_ (i.e., from 0 to 2000 μg/m^3^), (**c**) relative humidity, (**d**) temperature. Abbreviation: PM_2.5_, particulate matter with an aerodynamic diameter less than 2.5 μm. ^a^ Nineteen samples were from inside and four samples were from outside the homes (for PM_2.5_, 17 samples were from inside and three samples were from outside the homes).

**Table 1 ijerph-17-05681-t001:** Characteristics of homes at a brick kiln in Bhaktapur, Nepal, May 2019.

Characteristic	Homes, n (%)	Missing, n	Mean	SD	Min	Q1	Median	Q3	Max
Total	19 (100)								
Home area, m^2^			11.30	5.63	5.41	8.03	10.25	11.60	31.40
Number of people in home			3.42	1.43	1.00	2.00	4.00	4.00	7.00
Occupant density, number of people/100 m^2^			33.25	15.69	9.55	24.91	29.70	39.03	73.96
Number of children in home									
0-1	10 (53)								
2-3	9 (47)								
Smokers in home		2							
No	8 (47)								
Yes	9 (53)								
Number of smokers in home		2	0.76	1.03	0.00	0.00	1.00	1.00	4.00
Fuel type		1							
Gas	12 (67)								
Wood	6 (33)								

Abbreviations: Max, maximum; Min, minimum; Q1, first quartile; Q3, third quartile; SD, standard deviation.

**Table 2 ijerph-17-05681-t002:** Summary statistics for the mean of samples inside or outside ^a^ homes at a brick kiln in Bhaktapur, Nepal, May 2019.

Variable	Samples, n	Missing, n	Mean	SD	Min	Q1	Median	Q3	Max
PM_2.5_, μg/m^3^	20	3	272.13	389.54	19.37	58.73	118.46	278.49	1384.44
Relative humidity, %	23		57.87	6.37	46.52	52.66	58.39	60.47	73.70
Temperature, °C	23		24.58	1.66	20.66	23.72	24.88	25.89	26.87

Abbreviations: Max, maximum; Min, minimum; PM_2.5_, particulate matter with an aerodynamic diameter less than 2.5 μm; Q1, first quartile; Q3, third quartile; SD, standard deviation. ^a^ Nineteen samples were from inside and four samples were from outside the homes (for PM_2.5_, 17 samples were from inside and three samples were from outside the homes).

**Table 3 ijerph-17-05681-t003:** Associations between home characteristics and the mean of samples inside or outside^a^ homes at a brick kiln in Bhaktapur, Nepal, May 2019.

Characteristic	PM_2.5_, μg/m^3^			Relative Humidity, %			Temperature, °C		
	GM ^b^	95% CI ^b^	*p*-Value ^b^	Mean ^c^	95% CI ^c^	*p*-Value ^c^	Mean ^c^	95% CI ^c^	*p*-Value ^c^
Home area, m^2^	0.97 ^d^	0.87, 1.08 ^d^	0.54	0.01 ^e^	−0.37, 0.40 ^e^	0.95	−0.03 ^e^	−0.13, 0.07 ^e^	0.58
Number of people in home	0.88 ^d^	0.58, 1.35 ^d^	0.54	−1.04 ^e^	−2.47, 0.39 ^e^	0.14	0.01 ^e^	−0.39, 0.41 ^e^	0.96
Occupant density, number of people/100 m^2^	1.01 ^d^	0.97, 1.05 ^d^	0.48	−0.04 ^e^	−0.18, 0.10 ^e^	0.54	0.02 ^e^	−0.02, 0.05 ^e^	0.30
Number of children in home									
0–1	160.57	70.82, 364.06		57.13	54.40, 59.87		25.26	24.50, 26.02	
2–3	150.18	56.46, 399.50	0.91	54.21	51.32, 57.10	0.14	24.93	24.13, 25.73	0.54
Smokers in home									
No	61.65	36.05, 105.43		54.16	50.86, 57.45		24.58	23.82, 25.34	
Yes	301.35	176.20, 515.37	0.0005	56.91	53.80, 60.02	0.21	25.63	24.91, 26.35	0.05
Number of smokers in home	1.74 ^d^	1.08, 2.81^d^	0.03	1.19 ^e^	−1.10, 3.48 ^e^	0.29	0.55 ^e^	0.04, 1.06 ^e^	0.04
Fuel type and location									
Gas, indoor	79.32	49.82, 126.28		54.40	51.88, 56.92		24.67	24.04, 25.29	
Wood, indoor	541.14	288.31, 1015.68		58.51	54.94, 62.07		26.20	25.32, 27.08	
Outdoor	48.38	19.86, 117.87	<0.0001 ^f^	67.97	63.60, 72.33	<0.0001 ^g^	22.11	21.04, 23.19	<0.0001 ^h^

Abbreviations: CI, confidence interval; GM, geometric mean; PM_2.5_, particulate matter with an aerodynamic diameter less than 2.5 μm. ^a^ Nineteen samples were from inside and four samples were from outside the homes (for PM_2.5_, 17 samples were from inside and three samples were from outside the homes). Except for analyses involving fuel type and location, only indoor samples were used for analyses shown in this table. ^b^ Estimated via linear regression models of the natural logarithm transformed values. ^c^ Estimated via linear regression models of the original values. ^d^ Exponentiated regression coefficient and 95% CI (i.e., GM PM_2.5_ concentration ratio for a specified change in the independent variable or exp(β)−1 = percent change in GM PM_2.5_ concentration for a specified change in the independent variable). ^e^ Regression coefficient and 95% CI (i.e., change in relative humidity or temperature for a specified change in the independent variable). ^f^ Using the Tukey method to adjust for multiple comparisons, *p*-values for tests of pairwise differences among fuel type and location categories were as follows: gas, indoor vs. wood, indoor: 0.0002; gas, indoor vs. outdoor: 0.56; wood, indoor vs. outdoor: 0.0006. ^g^ Using the Tukey method to adjust for multiple comparisons, *p*-values for tests of pairwise differences among fuel type and location categories were as follows: gas, indoor vs. wood, indoor: 0.15; gas, indoor vs. outdoor: <0.0001; wood, indoor vs. outdoor: 0.006. h Using the Tukey method to adjust for multiple comparisons, *p*-values for tests of pairwise differences among fuel type and location categories were as follows: gas, indoor vs. wood, indoor: 0.02; gas, indoor vs. outdoor: 0.001; wood, indoor vs. outdoor: <0.0001.

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
