# Peer review of "Comparison of Liquefied Petroleum Gas Cookstoves and Wood Cooking Fires on PM2.5 Trends in Brick Workers’ Homes in Nepal"

_ijerph, 2020, doi:10.3390/ijerph17165681_

Round 1

Reviewer 1 Report

1. Line 30-32. Specify that it refers to open fires.
2. Line 148. Does the line refer only to exclusive users? What happens to people who use more than one fuel and more than one device?
3. Line 272-273. Specify that applies to the use of open fires, and not necessarily for chimney cookstoves and gasifiers that use firewood, briquettes, and pellets.
4. Line 285. The device-fuel relationship must be mentioned and not only fuel.
5. Line 318-325. An adoption or transition to LPG cannot be assumed. Must be monitored and reported.
6. Line 378. Future studies should include clean cooking stacking
7. The following points should be cited in the discussion section:
a. Is the price and supply of LP gas constant in the study area? Is there a government subsidy?
b. Are there only open fires and LP gas stoves in the study area? Can other fuels and technologies be incorporated in subsequent studies?
8. Add a section on study limitations.
a) This study does not consider the combined use of fuels and technologies. The impacts from the combined use of open fires and LP gas at the study site are unknown (cooking stacking).

Reviewer 2 Report

The authors examined the trends of LPG and wood cooking fire-related PM2.5 in Nepal. I am afraid that there is no novelty with the study, particularly in the context of environmental research and public health.

1) Firstly, the number of samples from convenient sampling where N = 19, is limited and thus limiting the interpretation of the results, and thereby leading to spurious results. Which is apparent in the wide intervals.

2) Likewise, the exposure was collected from April 30 to May 3, 2019, which renders even the effects estimate to be unstable due to the unaccounted temporal factors which confound the association.

3) Also, it was not clear what is the health outcome of interest. In Line 164, the authors mentioned "simple linear regression models to estimate unadjusted associations" of which exposure to which outcome?

Reviewer 3 Report

The Authors present a manuscript titled:" Comparison of liquefied petroleum gas cookstoves and wood cooking fires on PM2.5 trends in brick workers' homes in Nepal" interesting fo two major reasons: a) because indoor cooking and heating with biomass fuels including wood is recognized as a major source of PM2.5 exposure globally particularly among populations living below the poverty level  (as example for other similar conditions) b) because is the first study to characterize indoor PM2.5 levels in brick workers' homes in Nepal for an approximately 24-hour period .

The study design as well as the methodology used by Authors are fine and the statystical analysis appropriate and well led.

The limit of this study as recognized by Authors is the small number of homes at a single brick kiln in Bhaktapur, Nepal.The obtained results although significant could be not favourably accepted worldwide.

I don't have concern for the message of the manuscript and figures and tables are well inserted in the context of the paper I would like also to encourage Authors in the final part of Discussion and report in the Conclusions what they propose to overcome this problem. A not theoretical suggestion but considering also the poverty of some Countries what they thing could be done to avoid clinical respiratory damages.

Round 2

Reviewer 2 Report

This paper should be transferred in more exposure science journals.

Author Response

Reviewer comment:  This paper should be transferred in more exposure science journals.

Response:  We thank the reviewer for their thoughtful evaluation of our paper. In consideration of this comment, we have carefully reviewed the aims and scope of IJERPH. Based on the aims and scope of the journal, it is our opinion that IJERPH is a good fit for this paper for the following reasons: 

(1) IJERPH "focuses on the publication of scientific and technical information on the...interrelationships between environmental health and quality of life, as well as the socio-cultural, political, economic, and legal considerations related to...public health". In this paper, we report on the interrelationship between fuel/device type and indoor air quality in brick workers' homes. Our findings clearly show an interrelationship between cooking practices (which are heavily influenced by socio-cultural and economic factors) and indoor PM2.5 exposures in this population of workers. Thus, the paper fits with the intended purpose of the journal. 

(2) Our paper fits within the interdisciplinary scope of IJERPH, and addresses a critical environmental issue.  The aims and scope of IJERPH state that the journal "...links several scientific disciplines...to address critical issues related to environmental quality and public health." Exposure assessment represents one of many scientific disciplines working on this problem. Furthermore, our paper reports on a critical issue in global environmental health. Currently, the World Health Organization (WHO) considers household air pollution to be "the most important global environmental risk today" (WHO Guidelines for Indoor Air Quality: Household Fuel Combustion, pg. xiv).  While we agree with the reviewer that our paper is heavily focused on exposure assessment, we do not see in the aims and scope of the journal that submission of exposure assessment papers is discouraged. To the contrary, the aims and scope state that IJERPH is a comprehensive multidisciplinary journal. Accordingly, there are several air quality-related exposure assessment papers published in IJERPH in recent months, including: 

Evaluation on Air Purifier’s Performance in Reducing the Concentration of Fine Particulate Matter for Occupants according to its Operation Methods

by Hyungyu Park 1,Seonghyun Park 2OrcID andJanghoo Seo 3,*

Int. J. Environ. Res. Public Health 2020, 17(15), 5561; https://doi.org/10.3390/ijerph17155561

Household Exposure to Secondhand Smoke among Chinese Children: Status, Determinants, and Co-Exposures

by Muxing Xie 1,Chunrong Jia 2,*OrcID,Yawei Zhang 3OrcID,Beibei Wang 1,Ning Qin 1,Suzhen Cao 1,Liyun Zhao 4,Dongmei Yu 4 andXiaoli Duan 1,*OrcID

Int. J. Environ. Res. Public Health 2020, 17(15), 5524; https://doi.org/10.3390/ijerph17155524

Organic Air Quality Markers of Indoor and Outdoor PM2.5 Aerosols in Primary Schools from Barcelona

by Barend L. van Drooge 1,*OrcID,Ioar Rivas 1,2OrcID,Xavier Querol 1,Jordi Sunyer 2OrcID andJoan O. Grimalt 1OrcID

Int. J. Environ. Res. Public Health 2020, 17(10), 3685; https://doi.org/10.3390/ijerph17103685

Assessing the Impact of Housing Features and Environmental Factors on Home Indoor Radon Concentration Levels on the Navajo Nation

by Sheldwin A. Yazzie 1,*OrcID,Scott Davis 2,Noah Seixas 3 andMichael G. Yost 3

Int. J. Environ. Res. Public Health 2020, 17(8), 2813; https://doi.org/10.3390/ijerph17082813

(3)  Our paper fits into at least four of the 26 major sections of IJERPH, including Environmental Health, Global Health, Occupational Safety and Health, and Public Health Statistics and Risk Assessment (please note that the U.S. Environmental Protection Agency [2017] considers exposure assessment to be one of the main steps of conducting a human health risk assessment). 

We thank the reviewer again for their time in reviewing our manuscript.